**Data Availability Statement:** The data underlying the results presented in the study are available from the institutional representative, Ms. Pinko YQ

# Exploring the effects of web-based psychological capitals training on teachers' PsyCap development, emotional stability, and support: Evidence from Chinese inclusive education

**Yin Ru SHI**[1]*, **Kuen Fung Kenneth SIN**[2]

**1** Department of Special Education, Lingnan Normal University, Zhanjiang, Guangdong, China,
**2** Department of Special Education and Counselling, The Education University of Hong Kong, New Territory, Hong Kong, China

\* Shiyinru921020@163.com

## Abstract

Inclusive education for students with autism spectrum disorders (ASD) has increasingly received attention nationwide in China. Schools realize that teachers are under stress and lack the skills to handle daily interactions with these students. So far, few studies have directed efforts to provide a remedy for teachers to improve their daily work. This study aimed to design and implement a 2-hour web-based training on psychological capital (Psy-Cap) to protect their well-being and foster their supportive behavior for ASD students in the inclusive class. A total of 120 targeted teachers were invited to participate in the training and were randomly divided into control and treatment groups. Pre-, post-, and follow-up surveys were distributed before, after, and one week after the training. ANOVA results suggested that teachers showed a significant increase in their PsyCap scores after completing the training, although the training effect slightly decreased after one week. Moreover, higher levels of PsyCap showed a positive influence on teachers' emotional stability and supportive behavior in class. The results highlighted the effectiveness of web-based PsyCap training in boosting teacher positivity, which enhances teacher support for students in inclusive education.

## Introduction

Inclusive education receives a growing concern from international communities as increasingly more educational reforms have taken place in both developing and developed regions for a heightened understanding of education as a human right [1]. Despite this growing concern, the practical implementation of inclusive education encounters significant challenges, particularly in accommodating the varied developmental, cultural, socioeconomic, and familial backgrounds of students, including those with Autism Spectrum Disorders (ASD) [2]. Among these diverse student populations, children with ASD stand out as a minority group that

Wang. Her email address: pinkowang1989@gamil.com. She saved all the raw data collected in China in a hard disk for my research.

**Funding:** The authors would like to acknowledge the funding support provided by the Department of Education of Guangdong Province, China, through the Special Innovation Project for Guangdong Provincial Higher Education Institutions in 2022. The project number is 2022WQNCX042. The funders had no role in study design, data collection and analysis, decision to publish, or preparation of the manuscript.

**Competing interests:** The authors have declared that no competing interests exist.

demands considerable attention due to the complexities inherent in meeting their educational and social needs [3]. In this context, teachers emerge as critical agents for the successful enactment of inclusive practices, tasked with the dual responsibility of facilitating an inclusive classroom environment while addressing the special needs of each student [4, 5].

However, a notable gap in the literature reveals that teachers frequently feel not adequately equipped for this role, particularly in relation to effectively integrating children with ASD into their classrooms [6]. This gap is exacerbated by the pressures of educational reforms, which impose heightened standards and accountability, further compounded by challenges such as student misconduct and the demand for individualized care [7, 8]. These factors collectively contribute to increased stress, pessimism, and a decline in motivation among teachers, negatively impacting their overall wellbeing and job satisfaction [9]. In the quest for solutions to these challenges, the concept of Psychological Capitals (PsyCap) emerges as a promising avenue for research and practice [10]. PsyCap, with its components of hope, optimism, resilience, and self-efficacy, is posited as the vital psychological resources for enhancing individual performance, cognitive functioning, and motivation towards success [11, 12]. Despite the established benefits of PsyCap in various organizational settings, its application within the context of inclusive education, particularly as a support for teachers facing the multifaceted challenges of integrating ASD students, remains underexplored.

This study aims to bridge this critical gap by examining the potential of PsyCap to empower teachers within inclusive educational settings. Specifically, it seeks to design and evaluate a PsyCap training intervention in enhancing teachers' psychological resources, thereby facilitating a more positive and effective teaching environment for students with ASD. The significance of this research lies in its potential to offer evidence-based strategies for improving teacher preparedness and resilience, contributing to the broader goals of inclusive education by ensuring that teachers are well-equipped to meet the diverse needs of all students [5, 9]. Past studies underscore the transformative potential of PsyCap, with studies indicating its negative correlation with undesirable employee attitudes and behaviors, and a positive association with beneficial outcomes such as job satisfaction, performance, and well-being [13–15].

By focusing on the development and implementation of an online PsyCap training program, inspired by the pioneering work of Luthans and his associates [16, 17], this study tries to showcase how the PsyCap training can bolster the psychological resources of teachers. Prior investigations have demonstrated the feasibility and effectiveness of developing PsyCap through brief, technology-facilitated interventions [17]. For instance, Luthans et al. [17] illustrated that an intensive 2-hour web-based training could significantly enhance university students' PsyCap, leading to notable improvements in self-efficacy, optimism, and hope. Drawing upon these findings, the current study proposes a similar online training tailored for teachers in inclusive settings, aiming to equip them with the necessary psychological tools to address the challenges and stressors inherent in their roles. Building upon these foundations, this study not only addresses the existing research gap but also proposes a practical intervention that can potentially enhance the inclusivity and effectiveness of educational practices, thereby supporting the professional development of teachers and the academic success of students with ASD. This study answers the following research questions:

RQ1: Can teachers' PsyCap be effectively developed through the short web-based PsyCap training?

RQ2: Can this PsyCap training show sustained positive effects on teachers' PsyCap to be developed?

RQ3: How effective can this PsyCap training on teachers' sustained well-being and support to students?

## Literature review

### Theoretical background

In China, the concept of inclusive education emerged in the late 1980s, and since then, it has gained increasing attention from policymakers and educators. In 2010, the Chinese government launched the "Action Plan for the Development of Special Education (2010–2020)," which aimed to promote inclusive education and improve the quality of special education [18]. In response to the call for the equal right to education, the Chinese government has set a primary goal to increase school enrolment and retention by integrating children with ASD into mainstream schools to achieve the universalization of compulsory education [19]. Despite government efforts, challenges persist. Insufficient resources, including a lack of assistive technology, adapted materials, and appropriate facilities, pose a challenge for learners with diverse needs [20]. Furthermore, a lack of awareness and understanding of inclusive education among the general public contributes to stigma and discrimination towards these learners. However, the primary challenge lies in the inadequate teacher training on inclusive education. Many Chinese teachers lack the necessary knowledge and skills to effectively teach diverse learners [18].

Inclusive education, which aims to ensure that all students have access to quality education regardless of their background, has put increasingly more stress on teachers' shoulders nowadays. In China, such concern is also highlighted by educators and school leaders [21]. Teachers reported heightened stress when implementing inclusive practice in regular classrooms. For example, Brackenreed [22] identified the most common stressors during inclusion, such as workload, time management, lack of general support and locus of control, and insufficient teacher preparation. In addition, Jennett et al. [23] warned that teachers might be especially at risk for burnout in inclusive education when they work with ASD students. Taking into account the meta-analysis of Aloe et al. [24], behavioral disturbances in day-to-day interaction with students result in teachers' emotional exhaustion. Feelings of helplessness and loss of control also exacerbate stress.

Increasingly more researchers advocated that teachers in such situations should be given more opportunities to cultivate positive organizational behavior (POB). Following this, Luthans [25] introduced POB as a means to apply positive psychology in the workplace. The call for a positive approach to psychology is not a new concept. Maslow [26] suggested that psychology was too focused on negative aspects and should balance it with positive constructs. Positive psychology emerged in the literature in 2000, introducing positive constructs such as hope, optimism, subjective well-being, and individual development [27]. POB is defined as positively oriented human resource strengths and psychological capacities that can be measured, developed, and managed for performance improvement in today's workplace [25]. It emphasizes positive constructs that are state-like and thus open to development. Thus, organizations can benefit from cultivating and leveraging the positive psychological resources of their employees, leading to improved performance and well-being. Research has shown that promoting POB among teachers leads to increased job performance and satisfaction, reduced stress and burnout, and improved student outcomes [28]. Furthermore, positive attitudes and behaviors among teachers have been found to be contagious and can positively impact students' attitudes and behaviors [29]. Thus, it is crucial for schools to put more effort in cultivating POB among their teachers to create a positive and inclusive learning environment.

To gain a comprehensive understanding of psychological resources in organizational settings, it is crucial to examine the theoretical foundations of POB and its specific construct of PsyCap. PsyCap is a psychological resource that has been mentioned in various fields, including economics, sociology, and organizational psychology, and is believed to promote personal growth and well-being, which in turn can lead to organizational growth [10, 30]. Researchers have identified PsyCap as a valuable resource for protecting individuals from work-related stress [12]. Individuals with PsyCap are thought to be more equipped to adapt to and cope with stressful events, which can result in greater emotional stability, optimism, and cheerfulness in the face of negative stressors at work [31]. These teachers are likely to be psychologically resourceful, demonstrating qualities such as hope, optimism, resilience, and efficacy, which enable them to adapt to changing demands and evaluate problems in a balanced way. Existing studies on teachers with high levels of PsyCap have demonstrated promising results, such as high levels of work engagement, well-being, and interpersonal harmony [32]. PsyCap may be particularly crucial for teachers in demanding situations. Research has consistently shown that teachers with higher levels of PsyCap tend to exhibit more supportive behaviors towards their students [13]. PsyCap enhances the teacher's ability to cope with stress, develop positive relationships with students, and promote student motivation and engagement [32]. Specifically, teachers with higher levels of PsyCap tend to have a greater sense of self-efficacy, which leads to a more proactive and goal-oriented approach to teaching, using creative and innovative teaching strategies to meet the diverse needs of their students [32]. Similarly, teachers with higher levels of PsyCap tend to show optimism and hope, contributing to more teacher support, because they are better able to manage stress and job demands, and maintain a positive and supportive attitude towards their students [33]. Therefore, empowering teachers with PsyCap can promote teachers' well-being and foster their support for better students' adaptation and performance in class.

## PsyCap with four components

PsyCap encompasses four components: self-efficacy, optimism, hope, and resilience (Fig 1). These components have been extensively studied in the literature and have significant implications for individuals' well-being and performance in educational settings [12]. Hope, as described by Luthans and Youssef [12], involves perceiving a goal, identifying pathways to achieve it, and being motivated to pursue it. It plays a crucial role in educational contexts by providing individuals with a sense of direction and purpose. Studies have shown that higher levels of hope are associated with greater academic motivation, engagement, and achievement [34]. Students with high levels of hope are more likely to set challenging goals, develop effective strategies, and persist in the face of obstacles. Self-efficacy, another component of PsyCap, refers to an individual's belief in their ability to succeed in specific tasks or situations. This belief influences individuals' choices, effort, and persistence in academic endeavors. Students with high self-efficacy are more likely to set challenging goals, exert greater effort, and demonstrate higher levels of academic performance [35]. Developing self-efficacy can be facilitated through various strategies, including mastery experiences, social modeling, and verbal persuasion. Providing students with opportunities to successfully accomplish tasks, exposing them to positive role models, and offering supportive feedback can contribute to the development of self-efficacy beliefs. Resilience, the ability to bounce back from setbacks and difficulties, is another vital component of PsyCap in educational contexts. Students often face academic challenges, such as exams, assignments, and academic stress, which can impact their well-being and academic performance. Resilience helps students effectively cope with adversity and maintain their psychological well-being. Research has shown that resilient individuals exhibit higher

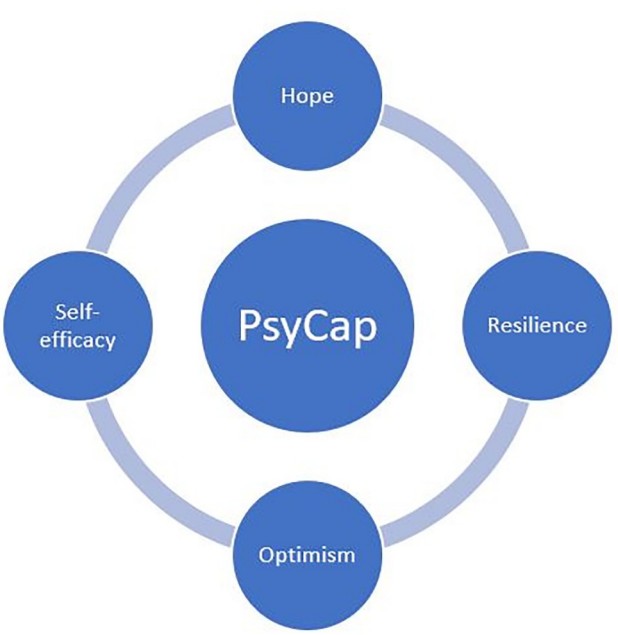

**Fig 1. PsyCap and its four components.**

levels of psychological well-being, adaptive coping strategies, and academic success [36]. Developing resilience can be supported through interventions focused on stress management, emotion regulation, and coping skills training. Optimism, the fourth component of PsyCap, involves the expectation of positive outcomes. Optimistic individuals tend to have a positive outlook on the future, viewing challenges as opportunities for growth and improvement. In educational settings, optimism has been linked to higher levels of academic motivation, engagement, and achievement [37]. Cognitive restructuring, a technique that involves challenging negative thoughts and replacing them with positive and realistic ones, and positive self-talk are effective strategies to develop and enhance optimism.

It is important to note that PsyCap is not a fixed trait but can be developed and strengthened through training interventions. PsyCap training provides individuals with a range of tools and techniques to enhance their positive psychological resources, ultimately contributing to their success in educational settings. By focusing on the development of self-efficacy, hope, resilience, and optimism, individuals can acquire the necessary skills and mindset to navigate challenges, overcome obstacles, and thrive academically.

### Emotional wellbeing

Teachers play a crucial role in ensuring the success of inclusive education, as they are responsible for creating a supportive and inclusive learning environment that meets the needs of diverse learners. However, the demands and challenges of inclusive education can affect teachers' emotional wellbeing, which can have negative consequences for both teachers and students. Inclusive education is associated with increased demands on teachers, such as managing diverse classrooms, adapting instruction to meet individual students' needs, and collaborating with other professionals and parents [38]. These demands can lead to emotional exhaustion, stress, and burnout, which can negatively impact teachers' wellbeing [39]. Research has shown that teachers who experience emotional exhaustion and burnout are more

likely to report lower job satisfaction, decreased commitment to teaching, and higher turnover rates [40]. In addition, teachers' emotional wellbeing can also impact student outcomes. A study by Jia et al. [41] found that teachers' emotional exhaustion negatively impacted student achievement, while teachers' emotional intelligence positively predicted academic performance. In addition, teachers' job satisfaction was positively associated with students' academic achievement and motivation. To support teachers' emotional wellbeing in inclusive education settings, various strategies have been proposed in the literature. For example, self-care strategies, such as mindfulness practices and physical exercise, can help teachers manage their stress and improve their emotional wellbeing [40]. In addition, providing teachers with professional development opportunities to enhance their knowledge and skills in inclusive education can be a good means to improving their confidence and reducing their stress [39].

## Teacher support

In inclusive education, teachers are required to provide social support to students to meet their unique needs. According to Anderson et al. [42], social support is essential for successful inclusive education. Social support is a multifaceted construct that refers to the provision of emotional, instrumental, informational, and appraisal assistance from others. Emotional support involves providing comfort, empathy, and understanding, while instrumental support involves providing practical assistance, such as transportation, financial support, or physical care. Informational support involves providing guidance, advice, or information, while appraisal support involves providing feedback, encouragement, and affirmation [43]. Teachers who provide social support can help students with special needs feel valued, respected, and included in the classroom, which can improve their academic and social outcomes. One study conducted by Tzuriel and Shamir [44] found that social support from teachers was positively related to students' academic achievement and social integration in inclusive education settings. Additionally, other studies have found that teachers who provide social support can reduce students' anxiety and stress levels and increase their self-esteem and self-efficacy [42]. However, providing social support in inclusive education can be challenging for teachers. According to Lopata et al. [45], teachers in inclusive education settings have reported that they are overwhelmed, stressed, and underprepared to meet the unique needs of their students. They also reported the lack of training in necessary knowledge and skills to provide social support in inclusive class.

## Web-based PsyCap training design

Insufficient teacher training in inclusive context becomes the critical hurdle for teachers to carry out classroom practice smoothly [46, 47]. Teachers may have limited knowledge about how to accommodate the unique characteristics of children with ASD to pedagogical adaptations [23]. For example, teachers who lack expertise show a hard time applying specialist knowledge, tailoring students' special needs, and coping with stressful events in inclusive classrooms [6]. To better provide equal educational opportunities for all learners, including those with diverse needs, well-trained and competent teachers are urgently needed. Thus, professional development is desirable as it offers a constructive, systematic approach to supporting teachers in fostering a sense of efficacy and competence by providing them with the necessary knowledge and skills to teach diverse learners effectively [48].

Teachers' professional development on PsyCap typically involves a series of workshops or sessions that focus on building individuals' positive psychological resources, such as self-efficacy, optimism, hope, and resilience. The design of PsyCap Training varies depending on the specific program, but most programs follow the design framework. (1) Assessment: before

starting the training, participants may complete assessments of their current levels of PsyCap. (2) Education: participants learn about the concepts of self-efficacy, optimism, hope, and resilience, including the benefits of each for personal and professional development. (3) Skill-building: participants engage in activities and exercises designed to develop and enhance their PsyCap. These may include goal-setting, cognitive restructuring, visualization techniques, and stress management strategies. (4) Reinforcement: positive feedback and reinforcement are used to reinforce positive behaviors and attitudes. (5) Maintenance: follow-up sessions or activities may be provided to help participants maintain the gains made during the training.

Past studies suggested that the use of online intervention technology has several advantages over traditional face-to-face interventions. PsyCap training programs have also been modified with technology-based features to accommodate the demands for online training [49, 50]. Luthans et al. [17] supported the use of the online PsyCap training intervention as their meta-analytic results indicated that web-based instruction might be as effective, or even more effective than traditional face-to-face classroom instruction. Online interventions are easily accessible to a large number of people, can be completed at any time or place, and are often more cost-effective than in-person interventions. In addition, online interventions may be particularly beneficial for individuals who are hesitant to seek help due to social stigma or other barriers. Online training with technology supports such as personalized animation, detailed PowerPoints, personalized exercises, and video commentary by a facilitator can maximize the learning and development of PsyCap [17].

Web-based interventions consisting of multiple modules aimed at developing self-efficacy, hope, resilience, and optimism, have been used to deliver PsyCap Training. For example, the study by Sumalrot et al. [50] aimed to evaluate the effectiveness of a web-based PsyCap intervention on the mental well-being of tourism workers during the COVID-19 pandemic. The intervention was designed to enhance self-efficacy, hope, and resilience over a 6-week period through various online activities. A randomized controlled trial was conducted, and the intervention group demonstrated significant improvements in their overall mental health and well-being compared to the control group. The study implied that the web-based delivery of the PsyCap intervention provided a flexible and accessible approach that can be adapted to various contexts. Therefore, organizations can implement similar interventions to support the mental health of their employees. Overall, the existing literature suggested that online PsyCap Training can be an effective way to develop and enhance the positive psychological resources of self-efficacy, optimism, hope, and resilience, leading to benefits such as increased job satisfaction, performance, well-being [10, 13] and reducing symptoms of depression and anxiety [51].

## Method

### Sample participants

Recruitment of the target teachers started on April 20th, 2020. In this study, 178 teachers teaching inclusive classes in a coastal city of China were invited. Target teachers were contacted through personal email for participation consent in the online "Positive Quality Teacher Training" project. All the participants provided informed consent after receiving a comprehensive explanation of the research aims and procedure details, emphasizing their right to withdraw at any time without consequences. They were ensured privacy and confidentiality through stringent anonymization and secure storage practices. In this study, the PsyCap training was designed to minimize any potential harm to participants, ensuring they experienced no physical, psychological, or emotional distress. In total, 120 of them agreed to participate and included for data analysis (response rate = .67). Most of the teachers were female (90.1%). Their age ranged from 21 to 50, with 47.6% between 21 and 30, 32.2% between 31 and 40, and

2.1 percent missing. Most of them (84.7%) received a university education. Their work tenure ranged from 1 year to 30 years (Mean = 15.55, SD = 8.67). The average age of the students they taught was 8.6 years old (SD = 2.31).

## Data collection

This study used a pre- and post-test quasi-experimental design. This design was chosen primarily for its strength in evaluating the effects of an intervention when random assignment might not be feasible due to logistical considerations. It allows us to observe the changes in the dependent variable before and after the intervention in the same group of subjects, thereby providing a clearer picture of the intervention's impact. The sample size for the treatment group included 60 target teachers, and the control group contained 60 teachers. The teachers were randomly assigned into the two groups. Teachers in the treatment group were provided with the online PsyCap Training, while teachers in the control group were not given the training, but had a decision-making exercise instead.

The online pre- and post-surveys were developed in a Chinese data collection platform. The target teachers were asked to create their own unique ID with a combination of their surname and the last four digits of their phone number, such that their pre- and post-surveys could be matched after submission. These teachers completed their demographic information and baseline PsyCap (T1) one week before they were randomly assigned to either a 2-hour PsyCap teacher training or a decision-making exercise. Both treatment and control groups received a post-survey on PsyCap, emotional stability, and teacher support upon completion of the intervention (T2). After a week, the teachers in the treatment group were further invited for a follow-up survey (T3) on these key variables.

## Intervention procedures

Based on the web-based PsyCap training design proposed by Luthans et al. [17], this current study also emphasized the effectiveness of personalized animation, detailed PowerPoints, personalized exercises, and video commentary by a facilitator in enhancing learning and development of PsyCap. The personalized animation and video commentary provide visual aids that can help the learners better understand the concepts and make the learning process more engaging. The detailed PowerPoints provide a structure for the learners to follow and provide clear and concise information on the topic being covered. The personalized exercises provide a practical application of the concepts, which can help learners to internalize and retain the information.

In this study, the teachers were provided a 2-hour web-based training session on PsyCap. The implementation of the intervention for the treatment group included two online sessions. The first session included a one-hour narrated interactive PowerPoint presentation on PsyCap training embedded in flash animation, with short video clips from popular movies used as examples of self-efficacy, hope, optimism, and resilience in dramatized settings. The second session included 50 minutes of personalized exercises in developing self-efficacy, hope, optimism, and resilience. In the second session, for the development of the four components of PsyCap, the teachers were asked to write an interactive teaching plan for their inclusive students for the coming class. They were asked to write down the importance of personal values and setting realistically challenging and personally valuable goals to complete the teaching plan. The teachers were directed to take the goals that were realistically challenging and break them down into smaller goals. This process of dividing large goals into smaller more manageable ones was also designed to increase the ability of hope. For the development of self-efficacy, Bandura [35] suggested that mastery experiences can help build self-efficacy. Mastery

experiences are based on direct, personal experience rather than secondhand accounts, which allow us to observe direct links between effort and performance, thereby increasing expectancy judgments about the ability to perform well in particular situations. Previously, the teachers were asked to start with small challenges and gradually increase the difficulty level during the teaching plan design. They were also asked to recall other successful colleagues who had done similar tasks, and how their colleagues feel and react when receiving constructive feedback from their students. Moreover, the teachers were also taught the tactic of self-persuasion, such that they could provide positive feedback and encouragement to motivate themselves to exert greater effort and ability to achieve their teaching plans in the inclusive class. The increased positive expectations about those outcomes were intended to contribute to developing optimism for achieving success. In addition, when the teachers practiced developing strategies to attain personal goals, negative expectations may be reduced, and thus, positively influence optimism [51]. While the teachers were doing the teaching plan task, they were taught the importance of developing resilience of PsyCap. Particularly, they were introduced the tactics such as facing reality, creating meaning in life, and improvising to find new ways to reach goals in building resilience to build up a better basis for enduring hardship.

## Measures

**Psychological capitals.**  A 12-item scale short version of PsyCap questionnaire [17] was adopted from the original 24-item PsyCap questionnaire [52]. In this study, the 12-item short questionnaire included 3 items for self-efficacy, hope, optimism, and resilience respectively. To emphasize the "state-like" nature of the measure, the participants were asked to respond by describing "how you may think about yourself right now." Each PsyCap component demonstrated acceptable reliability in this study from T1 to T3 (efficacy = .871-.890, hope = .761-.825, resilience = .865-.901, optimism = .760-.804), as well as overall PsyCap (.923-.943). Although psychometric support for the construct validity of PsyCap was demonstrated in many past PsyCap research [51], limited evidence of validity was reported in the inclusive education context. Therefore, confirmatory factor analysis of PsyCap was conducted in this study while considering PsyCap as a second-order factor (see theoretical discussion of second-order structure in Luthans et al., [17]). All of the item loadings were significant ($p < .01$) on their respective latent factor as well as each component loading on the second-order factor PsyCap. Results of the CFA were as follows: CFI = .920-.958, TLI = .926-.945, RMSEA = .050-.054. CFA results suggested good fit for the second-order factor model of PsyCap. Unless otherwise mentioned, all scales used in this study were anchored on a 5-point Likert scale ranging from 1 (strongly disagree) to 5 (strongly agree).

**Teacher support.**  Based on the original 60-item child and adolescent social support scale (CASSS) developed by Malecki et al. [53], further modifications were made to create a shortened version consisting of 10 items. This modified scale aimed to assess teachers' support in inclusive education. The development of the shortened scale was based on existing research on the CASSS scale and its psychometric properties. By selecting a subset of the original 60 items from the CASSS scale that were most representative of teachers' support, the goal was to capture the essential aspects of this construct while reducing respondent fatigue and increasing response rates. The selection of items was guided by theoretical considerations and expert judgment in consultation with educational researchers in the inclusive education field. In this study, the internal consistency reliability of the 10-item scale was evaluated using Cronbach's alpha, which was calculated to be.961-.977 (T2 and T3). This high value suggests that the items in the scale were highly interrelated and exhibited strong internal consistency, indicating that the scale was reliable for measuring teachers' support in the inclusive education context.

**Table 1. Correlations of key study variables for the treatment and control groups in post-survey (N = 120).**

|  | Mean | SD | Skewness | Kurtosis | 1 | 2 | 3 | 4 | 5 | 6 | 7 | 8 | 9 |
|---|---|---|---|---|---|---|---|---|---|---|---|---|---|
| 1.Age | 1.908 | .290 | -2.866 | 6.320 | | | | | | | | | |
| 2.Gender | 1.744 | .767 | .476 | -1.150 | .160 | | | | | | | | |
| 3.Education | 2.542 | .721 | .126 | -.260 | -.362** | -.042 | | | | | | | |
| 4.Tenure | 9.921 | 8.139 | .862 | -.482 | .805** | .223* | -.388** | | | | | | |
| 5.Student in Class | 30.966 | 22.897 | 5.247 | 31.352 | .083 | .076 | -.119 | -.017 | | | | | |
| 6.Student Age | 5.263 | 1.842 | 2.711 | 9.753 | -.188* | -.065 | .219* | -.170 | -.050 | | | | |
| 7.Weekly Teaching (min) | 151.200 | 119.884 | 1.090 | .892 | .082 | -.160 | -.153 | .101 | -.096 | .099 | | | |
| 8.PsyCap | 51.392 | 7.701 | -.735 | -.490 | -.080 | .092 | .131 | -.017 | -.056 | .109 | -.022 | | |
| 9.Emotion Stability | 3.999 | .598 | .017 | -1.013 | -.068 | .123 | .158 | -.066 | .056 | .106 | -.028 | .659** | |
| 10.Teacher Support | 4.573 | .624 | -1.456 | .953 | -.095 | .139 | .162 | -.050 | -.062 | .080 | -.037 | .856** | .615** |

Note.

*stands for p < .05;

** stands for p < .01 (two-tailed)

**Emotional stability.** Chaturvedi and Chander [31] developed a 16-item emotional stability scale to measure individuals' emotional stability. The development of this scale was guided by existing theories and research on personality traits, emotional well-being, and psychological resilience to capture the multifaceted nature of emotional stability. The scale included items that assessed emotional regulation, self-control, resilience, and adaptability. These dimensions reflect the underlying constructs of emotional stability, which have been consistently linked to individuals' overall psychological well-being and positive life outcomes [54]. In this study, Cronbach's alpha coefficient was.861-.920 (T2 and T3), indicating strong internal consistency.

## Results

### Descriptive analysis

Table 1 shows the descriptive statistics, reliabilities, and Pearson's correlations of the key study variables using the post-survey sample. Results showed that PsyCap, emotional stability, and teacher support were positively and significantly correlated with each other. Data were checked for normality and outliners. A few cases of extreme outliers were treated as missing (e.g., age = 120).

### Effectiveness of the PsyCap training

To answer **Q1**: how this 2-hour web-based PsyCap training could effectively influence students' development of PsyCap using data from T1 and T2. Firstly, the independent sample T test was conducted for treatment and control groups to make sure that the teachers' baseline PsyCap (T1) was roughly similar across the groups. Results showed no significant differences of PsyCap between the control group (M = 40.933, SD = 3.755) and the treatment group (M = 40.917, SD = 3.369), t(118) = .026, p = .979. Then, the mixed ANOVA was conducte to explore the effect of PsyCap training with a 2 (condition: treatment vs. control) x 2 (time: pre-test vs. post-test) factorial quasi-experimental design among 120 participating teachers. The Shapiro-Wilk test indicated that the data were normally distributed, W = 0.92, p = 0.176. The Levene's test indicated that the assumption of homogeneity of variances was met, $F(1, 118) = 2.337$, p = .129. The results revealed a significant main effect of condition, $F(1, 118) = 90.688$, p < .001, $\eta^2 = .435$, indicating that the treatment group (M = 57.617, SD = 2.337) had

significantly higher PsyCap scores than the control group (M = 45.167, SD = 5.944). There was also a significant main effect of Time, $F(1, 118) = 910.711$, $p < .001$, $\eta^2 = .885$, indicating that teachers' PsyCap significantly increased from the pre-test (M = 40.925, SD = 3.505) to the post-test (M = 51.392, SD = 7.701) regardless of treatment condition. Additionally, the results also indicated a significant interaction effect (Fig 2) between condition and time, $F(1, 118) = 323.001$, $p < .001$, $\eta^2 = .732$. All the effect sizes for the main effects of condition, time, and their interaction were large ($\eta^2 = .435$, $\eta^2 = .885$, and $\eta^2 = .732$, respectively). Moreover, paired sample T tests were conducted for both control and treatment groups. Results showed a significant increase in PsyCap for both control($t(59) = -7.368$, $p < .001$) and treatment ($t(59) = -42.974$, $p < .001$) groups over time regardless treatment condition. Nevertheless, treatment group ($\Delta$diff = 16.700) showed much significant increase in PsyCap than the control group ($\Delta$diff = 4.233).

Furthermore, the mixed ANOVA was also texted for each specific components of PsyCap (i.e., self-efficacy, hope, resilience, and optimism) in order to examine how this PsyCap training could nurture the four components of PsyCap specifically. Results showed similar patterns, indicating significant main effects of condition and time, and significant interaction effects of condition x time on the four components of PsyCap, namely self-efficacy, hope, resilience, and

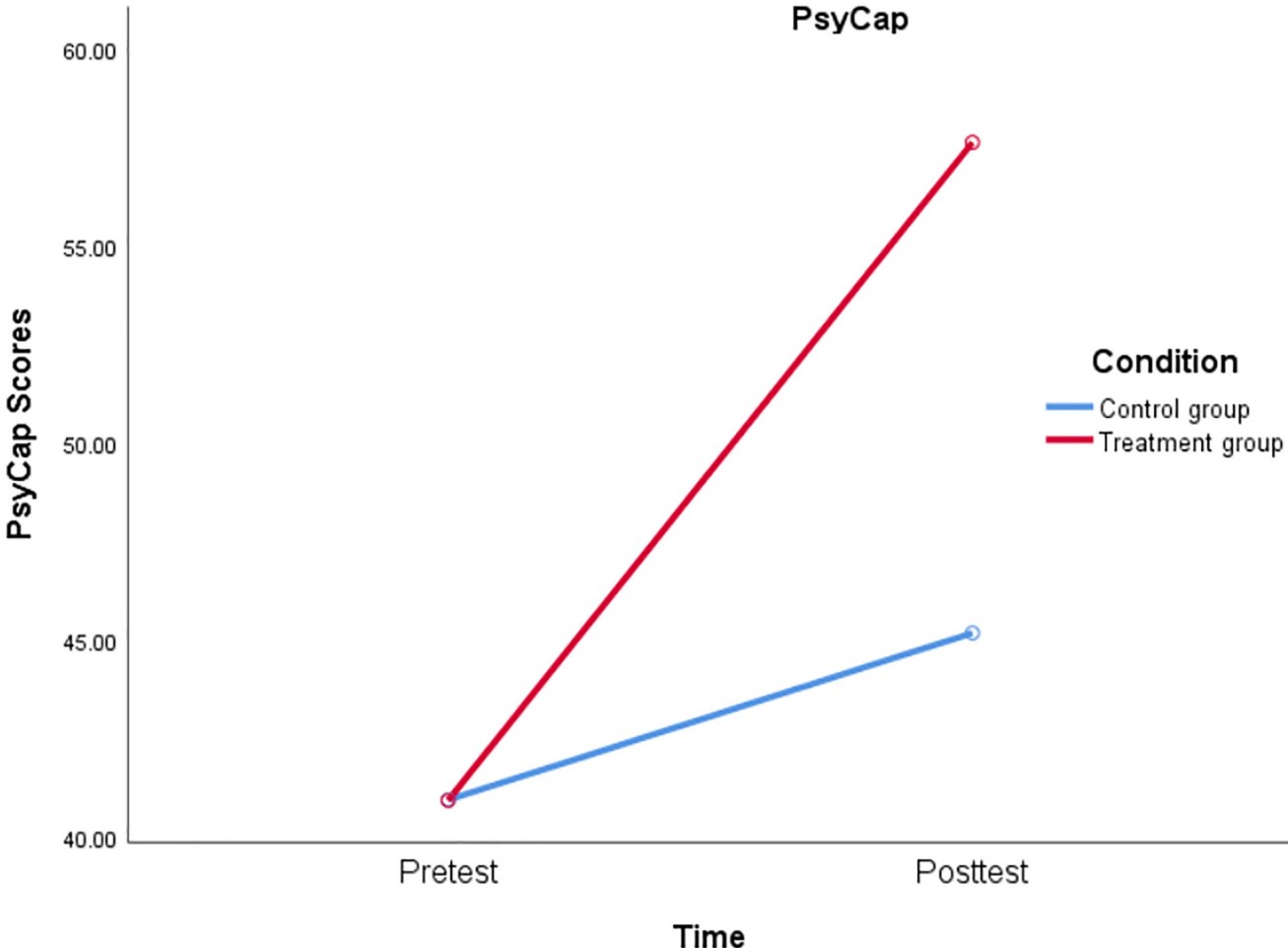

**Fig 2. Interaction plot of condition x time on PsyCap scores.**

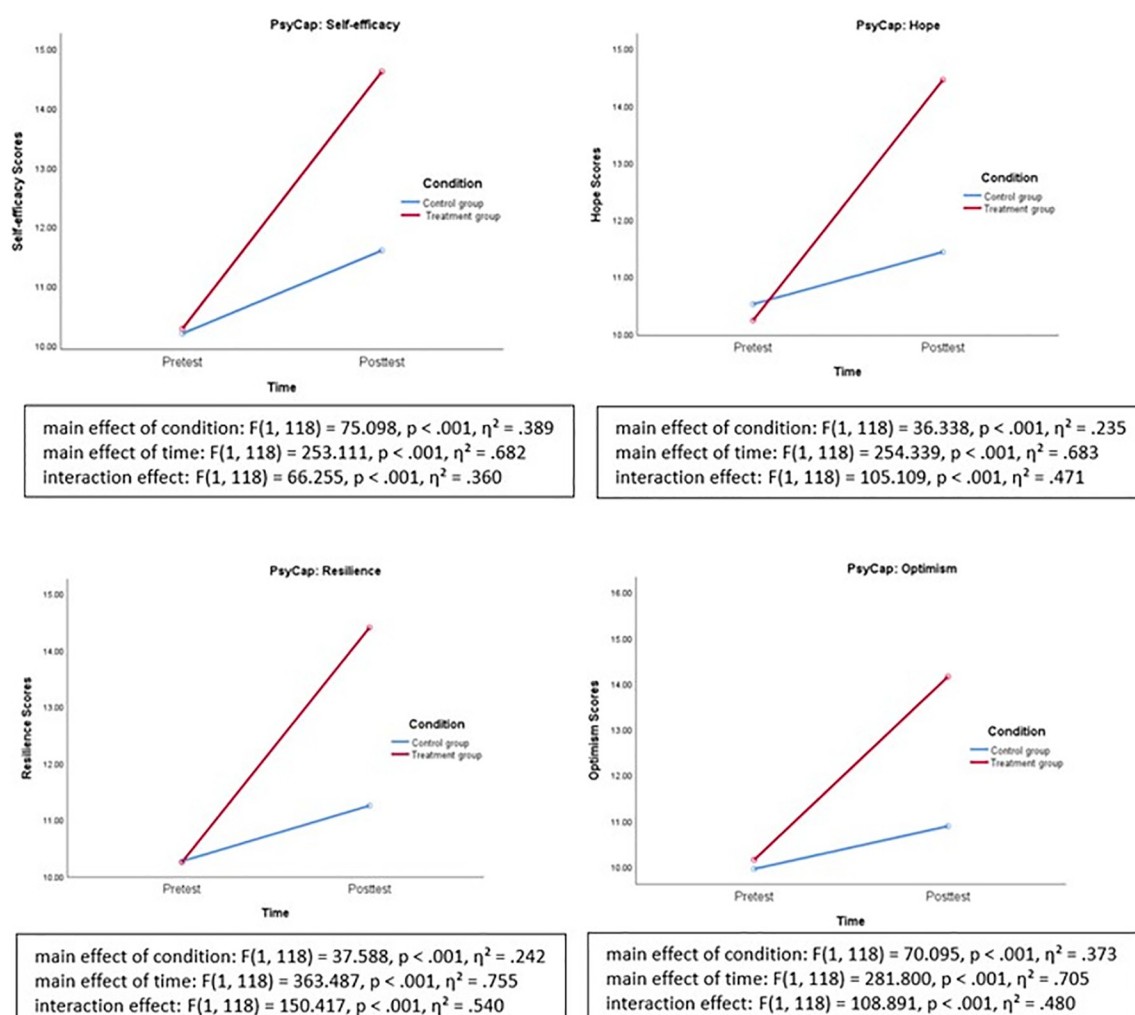

**Fig 3. Interaction plot of condition x time on the scores of the four components of PsyCap.**

optimism. The online PsyCap training seemed to have large effect sizes on the development of teachers' PsyCap, as well as its four components. Fig 3 shows the interaction plots for the four components of PsyCap.

### Sustainability of the PsyCap training

For **Q2**, the sustainability of the training effect was examined after completion of the training. Table 2 shows the Pearson's correlations of teachers' PsyCap from pre-test (T1), post-test(T2), and follow-up surveys (T3). One-way repeated measures ANOVA was conducted to investigate the impact of the web-based training on teachers' PsyCap before the training (pre-test), immediately after the training (post-test), and one week following the training (follow-up). Mauchly's test indicated that the assumption of sphericity had been violated, $\chi^2(2) = .507$, $p < .001$. Therefore, degrees of freedom were corrected using Greenhouse-Geisser estimates of sphericity ($\varepsilon = .670$). The results revealed a significant effect of time on PsyCap scores, $F(1.340, 118) = 1318.071$, $p < .001$, $\eta^2 = .957$. Post-hoc tests using the Bonferroni correction indicated that PsyCap scores significantly increased from pre-test (M = 37.500, SD = 3.011) to post-test

**Table 2. Correlations of PsyCap from T1 to T3 for treatment group sample (N = 60).**

| | 1 | 2 | 3 | 4 | 5 | 6 | 7 | 8 | 9 | 10 | 11 | 12 | 13 | 14 |
|---|---|---|---|---|---|---|---|---|---|---|---|---|---|---|
| **1.PsyCapT1** | | | | | | | | | | | | | | |
| **2.PsyCapT2** | .424** | | | | | | | | | | | | | |
| **3.PsyCapT3** | .384** | .786** | | | | | | | | | | | | |
| **4.EfficacyT1** | .408** | -.225 | -.137 | | | | | | | | | | | |
| **5.EfficacyT2** | .010 | .485** | .394** | -.178 | | | | | | | | | | |
| **6.EfficacyT3** | .043 | .288* | .432** | -.042 | .752** | | | | | | | | | |
| **7.HopeT1** | .616** | .455** | .352** | .163 | .060 | .127 | | | | | | | | |
| **8.HopeT2** | .275* | .534** | .490** | -.009 | .118 | .064 | .393** | | | | | | | |
| **9.HopeT3** | .306* | .445** | .430** | .009 | .006 | -.059 | .282* | .740** | | | | | | |
| **10.ResilienceT1** | .686** | .526** | .515** | -.011 | .217 | .175 | .268* | .259* | .225 | | | | | |
| **11.ResilienceT2** | .361** | .699** | .506** | -.107 | .041 | -.141 | .394** | .194 | .219 | .303* | | | | |
| **12.ResilienceT3** | .362** | .602** | .619** | -.056 | .062 | -.068 | .308* | .088 | .139 | .386** | .829** | | | |
| **13.OptimismT1** | .596** | .284* | .178 | -.117 | -.024 | -.100 | .187 | .085 | .148 | .284* | .296* | .217 | | |
| **14.OptimismT2** | .345** | .687** | .502** | -.242 | .094 | .099 | .255* | .194 | .142 | .534** | .353** | .347** | .255* | |
| **15.OptimismT3** | .237 | .424** | .669** | -.208 | .013 | -.006 | .071 | .273* | .249 | .437** | .226 | .238 | .178 | .535** |

Note.

*stands for p < .05;

** stands for p < .01 (two-tailed)

(M = 52.850, SD = 2.130), p < .001, and from pre-test to follow-up (M = 50.217, SD = 1.984), p < .001. However, the results showed a decrease in PsyCap scores from the post-test to follow-up (p < .001) after one week, even though the magnitude of decrease was much smaller than that of the increase during the training. The results provided further evidence that the web-based training significantly improved participants' PsyCap from pre-test to post-test, and yet these effects were gradually fading away over time (Fig 4).

## PsyCap and outcomes

**Q3** examined how effective this PsyCap training could contribute to sustained teachers' emotional stability in face of daily challenges with ASD students and increase their support after the training. Path analysis was conducted using the treatment group sample from T1 to T3. Specifically, teachers' demographic variables (T1) were included as controls on PsyCap (T2), and regressed emotional stability (T3) and teacher support (T3) on PsyCap (T2). The proposed path model provided an excellent fit to the data, $\chi^2(8) = 26.995$, CFI = .965, TLI = .955, RMSEA = .006. The results showed that PsyCap after controling teachers' gender, age, education, and tenure was significantly and positively related to emotional stability ($\beta = .051$ SE = .005, p < .001), and teacher support ($\beta = .069$, SE = .005, p < .001), indicating that PsyCap developed through the teacher training tended to contribute to higher levels of emotional stability and teacher support one week after completion of the training.

## Discussion

The primary aim of this study was to investigate the effectiveness of a web-based PsyCap training program tailored specifically for teachers engaged in inclusive education. The study objectives were to determine whether such a program could enhance key components of PsyCap—self-efficacy, hope, resilience, and optimism—and to establish the sustainability of these

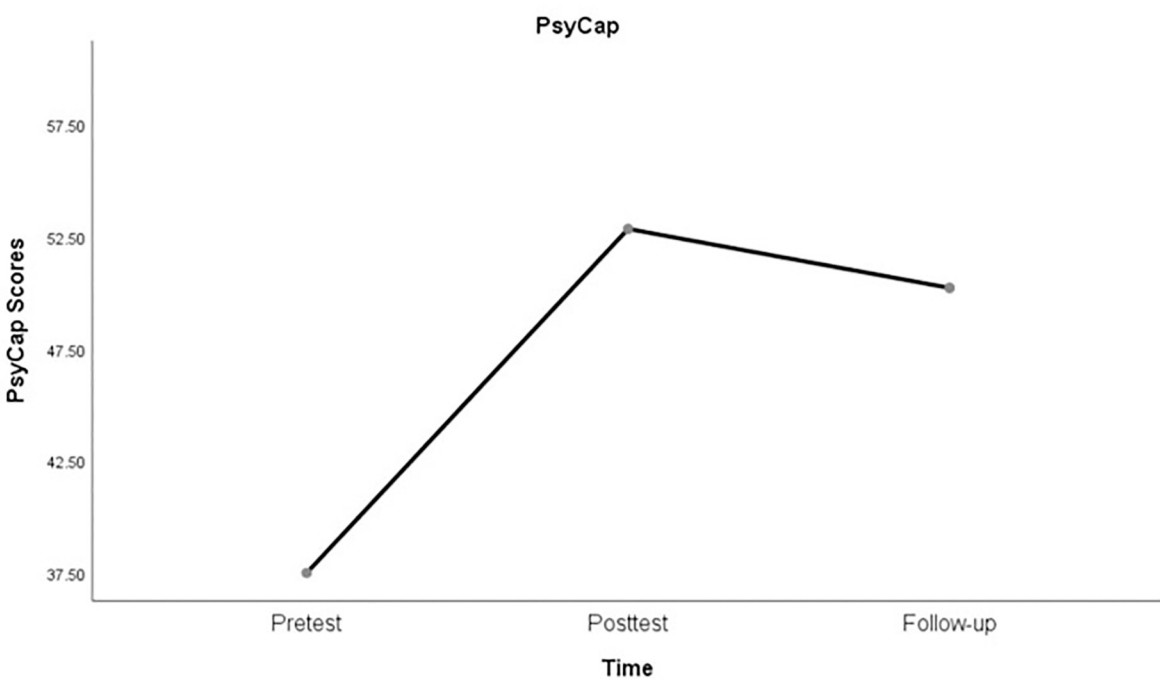

**Fig 4. The sustainability of the PsyCap training on teachers' PsyCap scores over time.**

enhancements over time. Furthermore, this study sought to explore the potential correlation between increased PsyCap among teachers and the quality of support provided to students, particularly those with additional educational needs.

The findings revealed that the 2-hour training was effective in enhancing teachers' PsyCap, including self-efficacy, hope, resilience, and optimism, as evidenced by large effect sizes in the development of these four components during the training. Such positive outcome resonates with the theoretical underpinnings of PsyCap posited by Luthans [25], suggesting that such psychological resources are crucial for performance improvement and stress reduction in the workplace. The sustainability of the training effects, evidenced by the maintained levels of Psy-Cap among teachers over time, addresses a critical gap highlighted in previous research [20, 23]. It suggests that the web-based training not only facilitates the initial development of these psychological capitals but also supports their sustainability, countering the traditional concerns regarding the ephemeral nature of training outcomes. This enduring effect is particularly noteworthy given the challenges of implementing inclusive education in China, such as insufficient resources and teacher preparation [18, 20]. Moreover, the observed positive correlation between enhanced PsyCap levels and improved emotional stability and support for students aligns with findings from Jennett et al. [23], who warned of the heightened risk of burnout among teachers in inclusive settings. This study contributes to the literature by demonstrating that enhancing teachers' psychological resources can mitigate such risks, fostering a more supportive and effective educational environment for students with additional needs.

The results of the study lend empirical support to the advocacy for cultivating PsyCap among teachers [25, 26], suggesting that such interventions can indeed lead to increased job satisfaction, reduced stress and burnout, and improved student outcomes [28]. The study findings were further supported by the teachers' feedback obtained from the open-ended questions

in the follow-up surveys, adding valuable qualitative evidence to scaffold the survey results. The feedback from teachers indicates the practical applicability and transformative potential of PsyCap training. This aligns with the broader educational goals of increasing school enrolment and retention by integrating children with ASD and other special needs into mainstream schools [19]. The majority highlighted the acquisition of new knowledge and skills for their well-being and work outcomes. One teacher mentioned, "What struck me the most was realizing that we have the ability to change things that were once considered unchangeable in our lives. . . It gives us hope to live positively and effectively cope with challenges in work and life." This sentiment reflects a shared recognition among teachers of PsyCap's benefits in fostering a positive workplace environment and enhancing their teaching practices. Some teachers also detailed how the training equipped them with strategies to boost confidence and manage classroom challenges more effectively. "During the training, we practiced techniques to enhance our confidence. . . I created learning profiles for these students to understand them better and provide more effective support when they face setbacks." Moreover, the training inspired a more supportive attitude in the classroom, with teachers acknowledging the role of optimism and hope in promoting supportive behavior towards students. "The training taught me to cultivate hope by setting goals and breaking them down into manageable steps. . . This coping strategy has proven to be very useful, especially in stressful situations, as regaining control ultimately brings about a sense of calm," reflected a teacher, emphasizing the utility of PsyCap in navigating the complexities of inclusive education.

## Contributions and implications

This study contributed to the inclusive education literature in several important ways. Firstly, it highlighted the critical role of fostering positivity in teachers through training, especially in challenging situations. Educators and school leaders have increasingly acknowledged the significant impact of teachers' positivity and well-being on their ability to effectively navigate and overcome various educational challenges [55]. The emphasis on nurturing positivity through training implies that equipping teachers with strategies and skills to develop a positive mindset can enhance their resilience and ability to cope with challenges such as student behavioral issues, workload demands, and limited resources.

Furthermore, fostering positivity in teachers not only benefits their own well-being but also positively influences student outcomes. Teachers who display positive emotions and attitudes are more likely to create a supportive and engaging classroom environment, which can lead to improved student motivation, learning outcomes, and well-being [56, 57]. Past research has underlined the importance of teacher well-being and a positive mindset in promoting student engagement, academic achievement, and a supportive classroom climate [29, 58, 59]. Thus, by providing teachers with the tools to develop and maintain a positive mindset, schools could improve teacher well-being, enhance classroom management effectiveness, and ultimately boost student outcomes.

Up till now, few studies have focused on implementing PsyCap training as part of professional development for teachers in Chinese inclusive education contexts. Sharma et al. [48] argued that professional development aimed at cultivating positive resources offers a constructive and systematic approach to support teachers. This approach helps in developing a sense of efficacy and competence by equipping them with the necessary knowledge and skills to teach diverse learners effectively. This study is among the few that have sought to nurture hope, optimism, self-efficacy, and resilience in teachers working with ASD children in China, emphasizing the development of PsyCap through training.

Moreover, this study makes a significant contribution to the field by designing and implementing a web-based PsyCap training program specifically tailored for teachers in inclusive settings. Web-based PsyCap intervention programs have gained considerable attention recently due to their potential to enhance PsyCap [17, 49, 50]. These programs use digital platforms to deliver targeted training and support, presenting several advantages over traditional face-to-face interventions. One major benefit of web-based interventions is their accessibility, allowing individuals to participate at their convenience and from different locations. This feature broadens the reach of interventions to a wider audience, including those who may not have access to in-person sessions due to geographical, logistical, or time constraints [60]. Another advantage of web-based interventions is their cost-effectiveness compared to traditional methods. By eliminating the need for physical infrastructure, travel, and instructor fees, these interventions become more affordable and accessible to organizations and individuals with limited resources [61]. The use of technology facilitates the scaling of PsyCap training, enabling a larger number of teachers to develop their psychological resources. This is crucial for combating workplace stressors and promoting well-being in challenging environments. It is because enhancing teachers' PsyCap can improve their awareness and strategies for supporting ASD students in class. Given the heterogeneity of ASD, teachers require diverse levels of support and care in their daily work [62]. PsyCap, therefore, emerges as a valuable psychological resource, protecting teachers from negative stressors and empowering them to provide tailored support in the inclusive education context.

## Limitation and future research directions

The study presented several limitations for future research effort. Firstly, the convenience sampling used in this study may have introduced selection bias, as participants who self-selected for the study might not represent the entire population of teachers. To enhance the generalizability of the findings, future research could employ more robust sampling techniques, such as random sampling or stratified sampling, to ensure a more representative sample of teachers [63]. Secondly, the reliance on self-rating measures raises the potential for social desirability bias, as participants may respond in a manner that aligns with societal norms or group expectations. To mitigate this bias, future studies could consider incorporating additional measures, such as observer ratings or behavioral assessments, to provide a more comprehensive and objective evaluation of teachers' experiences and behaviors [64]. Furthermore, the one-time-off nature of the PsyCap training in this study limited its long-term impact on teachers' behavior change. Research suggested that sustained and repeated interventions are more effective in producing enduring changes [65]. Future studies could explore the effectiveness of extended training programs, including longitudinal interventions or ongoing support, to examine the long-term intervention effects teachers' PsyCap development [66, 67].

## Supporting information

**S1 Data.**
(RAR)

## Author Contributions

**Conceptualization:** Yin Ru SHI.

**Data curation:** Yin Ru SHI.

**Formal analysis:** Yin Ru SHI.

**Funding acquisition:** Yin Ru SHI.

**Investigation:** Yin Ru SHI.

**Methodology:** Yin Ru SHI.

**Project administration:** Yin Ru SHI.

**Resources:** Kuen Fung Kenneth SIN.

**Software:** Yin Ru SHI.

**Supervision:** Kuen Fung Kenneth SIN.

**Validation:** Yin Ru SHI.

**Writing – original draft:** Yin Ru SHI.

**Writing – review & editing:** Kuen Fung Kenneth SIN.

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
