## [Decision Letter · Decision Letter 0]

2 Jan 2024

PONE-D-23-30803Exploring the Effects of Web-based Psychological Capitals Training on Teachers' PsyCap Development, Emotional Stability, and Support: Evidence from Chinese Inclusive EducationPLOS ONE

Dear Dr. SHI,

Thank you for submitting your manuscript to PLOS ONE. After careful consideration, we feel that it has merit but does not fully meet PLOS ONE’s publication criteria as it currently stands. Therefore, we invite you to submit a revised version of the manuscript that addresses the points raised during the review process.

We look forward to receiving your revised manuscript.

Kind regards,

Ali Derakhshan

Academic Editor

PLOS ONE

Journal Requirements:

"The authors would like to acknowledge the funding support provided by the Department of Education of Guangdong Province, China, through the Special Innovation Project for Guangdong Provincial Higher Education Institutions in 2022. The project number is 2022WQNCX042."

3. In this instance it seems there may be acceptable restrictions in place that prevent the public sharing of your minimal data. However, in line with our goal of ensuring long-term data availability to all interested researchers, PLOS’ Data Policy states that authors cannot be the sole named individuals responsible for ensuring data access (http://journals.plos.org/plosone/s/data-availability#loc-acceptable-data-sharing-methods).

6.Your ethics statement should only appear in the Methods section of your manuscript. If your ethics statement is written in any section besides the Methods, please move it to the Methods section and delete it from any other section. Please ensure that your ethics statement is included in your manuscript, as the ethics statement entered into the online submission form will not be published alongside your manuscript. 

6. Please include a separate caption for each figure in your manuscript.

**Additional Editor Comments:**

Dear Dr. Shi,

Thank you for your submission. Please revise your paper, highlight the changes, and provide a response letter.

You need to improve your theoretical framework and link your results to this framework. 

Best,

Ali Derakhshan

Reviewers' comments:

Reviewer's Responses to Questions

**Comments to the Author**

1. Is the manuscript technically sound, and do the data support the conclusions?

Reviewer #1: Yes

Reviewer #2: Yes

2. Has the statistical analysis been performed appropriately and rigorously? 

Reviewer #1: Yes

Reviewer #2: Yes

3. Have the authors made all data underlying the findings in their manuscript fully available?

Reviewer #1: Yes

Reviewer #2: Yes

4. Is the manuscript presented in an intelligible fashion and written in standard English?

Reviewer #1: Yes

Reviewer #2: Yes

5. Review Comments to the Author

Reviewer #1: Dear author(s),

Thanks for submitting your paper in the PLOS ONE journal. Overall, the present paper is a well-written paper that has a good potential to be accepted for publication. Howver, before any positive decision, the paper needs to be revised based on the comments included in the manuscript.

Best,

Reviewer #2: Dear Editor,

Thank you very much for offering me this valuable chance to review this illuminating article entitled “Exploring the Effects of Web-based Psychological Capitals Training on Teachers'

PsyCap Development, Emotional Stability, and Support: Evidence from Chinese

Inclusive Education”. The study clearly touches the theoretical and empirical bases of the variables separately and connects them logically. It truly adds to the current literature regarding students with autism spectrum disorders (ASD). Despite these merits, the study can improve by considering the following minor revisions. In case the next version is amended, the manuscript can be accepted.

1) In the abstract, revise the last sentence “Our findings highlighted the importance of nurturing teachers' PsyCap…..” Use the results highlighted……..

2) Add an introduction section to the paper (3-4 paragraphs). The current way of presenting information resembles LR. You need to follow “Introduction, LR, Method, Results, Discussion, Conclusion sections”.

3) The gaps can be further explained. Beyond lack of research, what gaps motivated you to run this study? Why is it significant?

4) You can add a figure to “PsyCap with Four Components”.

5) Some paragraphs are too long. Please, break them into shorter ones.

6) In the participants section, please add mean and SD of the participants’ age range and other background information. The readers need to know some demographics for replication.

7) Explain why have you chose this design and not others?

8) How did you ensure ethics of research? Which ethics did you observe in the study? Add them please.

9) In discussing your findings, try to connect to theoretical foundations mentioned in LR.

10) More implications and future directions can be added to the final part of the article.

11) Check the list of references based on the journal’s guidelines.

6. PLOS authors have the option to publish the peer review history of their article (what does this mean?). If published, this will include your full peer review and any attached files.

Reviewer #1: **Yes: **Mohammadsadegh Taghizadeh

Reviewer #2: **Yes: **Farhad Ghiasvand

---

## [Author Response · Author response to Decision Letter 0]

23 Feb 2024

Thank you for your positive feedback. We are grateful for the opportunity to refine our work further. We have carefully considered each point raised in your attachment and have formulated a detailed revision to address them comprehensively. We believe these changes will significantly improve our paper. For your convenience, we have listed your comments one by one with our corresponding responses. Please refer to the rebuttal letter attached in this revision and resubmission. Thank you!

---

## [Editor Report · Decision Letter 1]

26 Feb 2024

PONE-D-23-30803R1Exploring the Effects of Web-based Psychological Capitals Training on Teachers' PsyCap Development, Emotional Stability, and Support: Evidence from Chinese Inclusive EducationPLOS ONE

Dear Dr. SHI,

Thank you for submitting your manuscript to PLOS ONE. After careful consideration, we feel that it has merit but does not fully meet PLOS ONE’s publication criteria as it currently stands. Therefore, we invite you to submit a revised version of the manuscript that addresses the points raised during the review process.

We look forward to receiving your revised manuscript.

Kind regards,

Ali Derakhshan

Academic Editor

PLOS ONE

**Additional Editor Comments:**

Dear Dr. Shi,

Thank you for your submission. Please highlight the changes and write a detailed response letter.

Best,

Ali Derakhshan

---

## [Author Response · Author response to Decision Letter 1]

20 Mar 2024

I submitted the revised version of my manuscript through your online submission system, taking great care to adhere to the journal's submission requirements. This included providing (1) a manuscript marked with track changes, (2) a clean version of the revised manuscript, and (3) a detailed response letter addressing the reviewers' comments. Additionally, I ensured that all files were named according to the guidelines provided by your system.

---

## [Decision Letter · Decision Letter 2]

18 Jun 2024

Exploring the Effects of Web-based Psychological Capitals Training on Teachers' PsyCap Development, Emotional Stability, and Support: Evidence from Chinese Inclusive Education

PONE-D-23-30803R2

Dear Dr. SHI,

We’re pleased to inform you that your manuscript has been judged scientifically suitable for publication and will be formally accepted for publication once it meets all outstanding technical requirements.

Kind regards,

Ali Derakhshan

Academic Editor

PLOS ONE

Additional Editor Comments (optional):

Dear Dr. Shi,

Thank you for your submission.

Best,

Ali Derakhshan

Reviewers' comments:

Reviewer's Responses to Questions

**Comments to the Author**

1. If the authors have adequately addressed your comments raised in a previous round of review and you feel that this manuscript is now acceptable for publication, you may indicate that here to bypass the “Comments to the Author” section, enter your conflict of interest statement in the “Confidential to Editor” section, and submit your "Accept" recommendation.

Reviewer #2: All comments have been addressed

2. Is the manuscript technically sound, and do the data support the conclusions?

Reviewer #2: Yes

3. Has the statistical analysis been performed appropriately and rigorously? 

Reviewer #2: I Don't Know

4. Have the authors made all data underlying the findings in their manuscript fully available?

Reviewer #2: Yes

5. Is the manuscript presented in an intelligible fashion and written in standard English?

Reviewer #2: Yes

6. Review Comments to the Author

Reviewer #2: (No Response)

7. PLOS authors have the option to publish the peer review history of their article (what does this mean?). If published, this will include your full peer review and any attached files.

Reviewer #2: **Yes: **Farhad Ghiasvand

---

## [Editor Report · Acceptance letter]

8 Aug 2024

PONE-D-23-30803R2 

PLOS ONE

Dear Dr. SHI, 

I'm pleased to inform you that your manuscript has been deemed suitable for publication in PLOS ONE. Congratulations! Your manuscript is now being handed over to our production team.

Kind regards, 

on behalf of

Dr. Ali Derakhshan 

Academic Editor

PLOS ONE